# Ferroptosis and Iron Metabolism after Intracerebral Hemorrhage

**DOI:** 10.3390/cells12010090

**Published:** 2022-12-25

**Authors:** Yuanyuan Sun, Qian Li, Hongxiu Guo, Quanwei He

**Affiliations:** 1Department of Neurology, Union Hospital, Tongji Medical College, Huazhong University of Science and Technology, Wuhan 430022, China; 2Department of Rehabilitation, Tongji Hospital, Tongji Medical College, Huazhong University of Science and Technology, Wuhan 430030, China

**Keywords:** ferroptosis, intracerebral hemorrhage (ICH), iron metabolism, lipid peroxidation

## Abstract

The method of iron-dependent cell death known as ferroptosis is distinct from apoptosis. The suppression of ferroptosis after intracerebral hemorrhage (ICH) will effectively treat ICH and improve prognosis. This paper primarily summarizes the mechanism of ferroptosis after ICH, with an emphasis on lipid peroxidation, the antioxidant system, iron metabolism, and other pathways. In addition, regulatory targets and drug molecules were described. Although there has been some progress in the field of study, there are still numerous gaps. The mechanism by which non-heme iron enters neurons through the blood–brain barrier (BBB), the mitochondrial role in ferroptosis, and the specific mechanism by which lipid peroxidation induces ferroptosis remain unclear and require further study. In addition, the inhibitory effect of many drugs on ferroptosis after ICH has only been demonstrated in basic experiments and must be translated into clinical trials. In summary, research on ferroptosis following ICH will play an important role in the treatment of ICH.

## 1. Introduction

Intracerebral hemorrhage (ICH) is one of the most catastrophic events associated with high mortality and morbidity. The influx of blood ooze into the brain parenchyma form hematoma and lead to surrounding tissue oppression, which is the primary injury of ICH.

Injury to brain tissue following ICH includes two main categories: primary and secondary brain injury. The primary brain injury occurs within a few hours after the onset of ICH in which the initial hematoma causes the local brain parenchyma destruction from the mass effect [1]. Secondary brain injury, which is closely related to poor prognosis and is considered as a potential target for ICH treatment, is involved in the new brain tissue damage caused by the inflammatory reaction, blood-brain barrier (BBB) disruption, and perihematomal edema formation [2].

Unfortunately, after decades of hard work focusing on primary injury, there is no effective treatment to date, including for intensive blood pressure (BP) reduction, hemostasis, and hematoma evacuation. A series of Randomized controlled trials (RCTs), such as an Intensive blood pressure reduction in acute cerebral hemorrhage trial (INTERACT1, 2) [3], Antihypertensive treatment of acute cerebral hemorrhage trial (ATACH–I, II) [4], Factor seven for acute hemorrhagic stroke trial (FAST) [5], and Tranexamic acid in Intracerebral Hemorrhage (TICH) [6] have shown that intensive BP reduction or hemostasis might prevent the enlargement of hematoma but failed to reduce the mortality or major disability at 90 days. With regard to hematoma evacuation, Surgical trials in lobar intracerebral hemorrhage (STICH–I, II) [7,8] and Minimally Invasive Surgery and rt–PA for Intracerebral Hemorrhage Evacuation (MISTIE– I, III) [9] have shown that hematoma could be cleared, but the benefit for the long-term functional outcomes could not be found. Thus, we have to gaze at the secondary injury. The clot constituents, such as thrombin, complement, and cellular debris, can have toxicity, immunogenicity to the central nervous system, inducing oxidative stress and inflammatory response, and, finally, lead to various types of cell death, including necrosis, apoptosis, necroptosis, pyroptosis, and ferroptosis. Especially, hemoglobin in red blood cells generates neurotoxic reactive oxygen species that attack cell membranes, proteins, and DNA [10]. Excessive iron, disintegrated from hemoglobin after ICH, can induce a Fenton reaction and lead to ferroptosis.

Ferroptosis was first termed by Dixon in 2012 [11] and defined as a regulated cell death (RCD) induced by oxidative stress and regulated by glutathione peroxidase 4 (GPX4) by the Nomenclature Committee on Cell Death (NCCD). As a newly discovered form of RCD, ferroptosis is identified as shrinking mitochondria with an increased mitochondrial membrane density and breakdown of mitochondrial cristae, but a relatively intact nucleus in morphological features. In biochemical characteristics, iron accumulation and lipid peroxidation have been underlined before, and, at present, excessive oxidative stress and the inactivation of the cellular antioxidant system have been believed as the main causes of ferroptosis. Mounting evidence has shown that oxidative stress and ferroptosis may play a significant role in multiple physiological, such as embryonic development, and pathological processes [12]. Emerging evidence suggests that ferroptosis is also involved in neurological disorders, including stroke, degenerative diseases, neurotrauma, and cancer. The relationship between ICH and ferroptosis has attracted widespread attention.

In this review, we will briefly summarize the mechanisms of ferroptosis, and highlight the role of targeting molecular mediators of ferroptosis in ICH to shed a light on finding novel potential targets of ICH therapy.

## 2. The Mechanisms and Regulation of Ferroptosis after ICH

Spontaneous ICH is caused by bleeding into the brain parenchyma resulting from the rupture of blood vessels and cerebral small vessel disease (SVD), affected by a small arteriole, is the most prevalent cause of spontaneous ICH. Deep perforator arteriopathy and cerebral amyloid angiopathy (CAA) are the common types of sporadic SVD. A minority of spontaneous ICHs is caused by the rupture of large blood vessels, such as arteriovenous malformations and cavernous hemangiomas [13]. Free iron is released from degraded red blood cells (RBCs) and deposited around the hematoma within a few months. Through the Fenton or Haber–Weiss reactions, excessive free iron can generate hydroxyl radicals, leading to apoptosis and necrosis, gray matter damage, and brain–blood barrier (BBB) destruction [2]. Iron entering neurons can induce ferroptosis with lipid peroxidation as the core. The specific mechanism underlying ferroptosis is outlined below.

### 2.1. Iron Metabolism

Iron is an essential trace element that maintains the life of living organisms, involved in a variety of metabolic pathways, which is mainly present in the form of Fe^2+^ and Fe^3+^ in the human body. The body maintains a total iron level of 3–4 g, and the entire daily loss of iron is often just 1–2 mg. In general, iron is absorbed mostly through the diet in the form of Fe^3+^ and hemoglobin is where the majority of the body’s iron is found. Contrarily, plasma iron plays a role in the production of RBCs and is primarily coupled to transferrin (Tf). Adult males retain approximately 1g of ferritin in the liver as hepatocytes and macrophages [14]. Only the absorption of duodenal iron can regulate the amount of iron entering the body. At the apical brush border of the duodenal intestinal epithelium, duodenal cytochrome b (DCYTb) reduces Fe^3+^ to Fe^2+^, facilitating non-heme iron uptake via the divalent metal transporter 1 (DMT1), whereas the membrane iron transporter protein (FPN) exports iron through the basolateral membrane [15]. The process of iron absorption by enterocytes is mainly regulated by the systemic regulation of the hormone hepcidin and by intracellular iron regulatory proteins (IRPs). Hepcidin’s primary function is to regulate intestinal iron absorption by inducing the internalization and degradation of the FPN. When the body requires more iron for erythropoiesis or other processes, transferrin binds trivalent iron and transports it to the relevant organs for utilization or storage, although the precise transport mechanism is unknown [16].

As iron is required for proper brain metabolism, iron dysregulation impairs brain function. After ICH, erythrocyte red blood cell lysis can be observed, followed by the release of large quantities of potentially neurotoxic iron into the brain parenchyma, causing neurotoxicity and brain cell death [2]. Hematoma clearance after ICH is associated with macrophages/microglia and astrocytes. Macrophages are generally divided into two types, tissue-resident macrophages and blood-derived phagocytes. In brain tissue, tissue-resident macrophages are also known as microglia, and activated microglia release chemokines that recruit blood-derived macrophages to the hemorrhagic area. Activated phagocytes engulf deposited blood, damaged and dead tissue, which in turn provides a trophoblastic environment for local tissue reconstruction.

Microglia generally clear hematomas after ICH through three pathways: neuroglia bind to erythrocytes via CD36 to promote hematoma clearance; however, the overactivation of microglia causes the apoptosis of microglia and release of harmful iron [17]; extravasated RBCs that are not phagocytosed undergo free radical damage and complement-mediated spontaneous lysis, resulting in hemoglobin (Hb) release [18]. Haptoglobin (Hp) can form a stable Hb–Hp complex with hemoglobin (Hb), and microglia facilitate hematoma clearance by mediating the uptake of this complex through CD163 receptors [2]; Hb is also found to degrade intracellularly to the heme in hematoma and perihematomal brain tissue. Hemopexin (Hpx), a high-affinity heme-scavenging protein, can bind to the heme to form a stable heme–Hpx complex. The complex then undergoes receptor-mediated endocytosis via a CD91/low-density lipoprotein receptor-associated protein l (LRP1)[1,2].

Some studies have found that bexarotene administration facilitates recovery after ICH by enhancing hemophagocytosis, modulating macrophage phenotypes, and improving functional recovery [19]. Astrocytes produce inflammatory mediators to promote the transition from M1 to M2 microglia, and the limited studies available suggest that the interaction between microglia and astrocytes has not been fully investigated [20]. Further study of astrocyte mediators that regulate microglia polarization, phagocytosis, and other functions will improve our understanding of ICH pathology [21].

The heme that enters the cell by the above pathway is broken down by heme oxygenase–1 (HO–1) into iron, carbon monoxide (CO), and biliverdin, and the iron is transported extracellularly by the iron export protein FPN in microglia, causing a large amount of toxic iron to enter the brain parenchyma [22]. Extracellular iron binds to transferrin on neurons and enters the neurons, causing neuronal death. Two-molecule Fe^3+^ participates in the transport of iron by binding to one molecule of transferrin, which transports Fe^3+^ to the intracellular by binding to the membrane protein transferrin receptor 1 (TFR1) on the surface of neurons to form a Tf–Fe^3+^–TFR1 complex. Fe^3+^ is then reduced to Fe^2+^ by the Six-Transmembrane Epithelial Antigen of Prostate 3 (STEAP3). Endosomes release Fe^2+^ into the cytoplasmic labile iron pool (LIP), which requires DMT1/Solute Carrier Family 11 Member 2 (SLC11A2) regulation [23,24,25] (Figure 1). Fe^2+^ in the LIP can participate in the synthesis of the lipid peroxidation key enzyme lipoxygenases (LOXs), resulting in ferroptosis. In addition, Fe^2+^ will also participate in the Fenton reaction with hydrogen peroxide to produce hydroxyl radicals, attacking cellular components and causing ferroptosis. Fe^2+^ can also induce lipid auto-oxidation by producing reactive oxygen species (ROS) by iron-catalyzed enzymes, thereby promoting ferroptosis [23]. Ferritinophagy is also mediated by nuclear receptor coactivator 4 (NCOA4), which binds to ferritin and then transports iron-bound ferritin to the autophagosome for lysosomal degradation and iron release. NCOA4 knockdown can prevent lipid peroxidation and ferroptosis by reducing the amount of iron in the intracellular LIP [26]. Further to that, existing research indicates that the FPN is the only protein that transports iron out of cells, lowering the concentration of Fe^2+^ in cells and, thereby, inhibiting ferroptosis. In summary, iron plays an important role in the mechanism of ferroptosis after ICH, and future research into iron metabolism pathways may provide a new therapeutic avenue for alleviating brain injury after ICH.

### 2.2. The Lipid Peroxidation Pathway

Lipid peroxidation, a crucial mechanism that directly initiates ferroptosis, is the process by which oxygen binds to lipids to produce lipid peroxides by forming peroxyl radicals. The Fenton reaction resulting from iron stimulation produces lipid ROS, causing lipid peroxidation and eventually leading to ferroptosis [27]. In addition, the oxidation and esterification of polyunsaturated fatty acids (PUFAs) also produce lipid peroxides, which trigger lipid peroxidation [28]. According to earlier reports, AA–OOH–PE can induce ferroptosis [29]. Specific PUFAs containing phosphatidylethanolamines, including arachidonic acid (AA), are preferentially oxidized. Aryl–CoA synthase long–chain family 4 (ACSL4) catalyzes the conversion of exogenous AA to arachidonyl–CoA (AA–CoA), which is then esterified to arachidonyl–phosphatidylethanolamines (AA–PE) by lysophosphatidylcholine acyltransferase 3 (LPCAT3). Finally, AA–PE is then oxidized by LOXs to AA–OOH–PE, eventually leading to ferroptosis [28,30,31]. Neurons are vulnerable to lipid peroxidation because of the high concentration of PUFA glycerol phospholipids in the brain’s neuronal membrane [32]. However, further research is needed on the link between neuronal PUFAs and ferroptosis after ICH. Under physiological conditions, glutathione peroxidase 4 (GPX4) reduces AA–OOH–PE to AA–OH–PE to protect cell membranes from damage by toxic lipid peroxides. However, ferroptosis occurs when the content of AA–OOH–PE exceeds the ability of the reducing system [29]. Thus, ferroptosis can be prevented by reducing the production of reactive oxygen species and it is also possible to use ACSL4 and LOXs as targets for treatment [33,34,35] (Table 1).

The evidence suggests that cytochrome P450 oxidoreductase (POR), which may exist on the surface of the endoplasmic reticulum, can cause lipid leakage of lipid membranes, resulting in the production of lipid peroxidation, which in turn causes ferroptosis in cells [46,47]. Additionally, studies have demonstrated that POR and NADH–cytochrome b5 reductase 1 (CYB5R1) cooperate to promote lipid leakage in the lipid membrane and that the suppression of POR’s downstream targets has little impact on lipid leakage. However, it is still possible that some or all of the POR’s downstream elements could have an impact on lipid peroxidation [46]. Additionally, there is no proof that this is connected to the specific mechanism of lipid peroxidation in ferroptosis following ICH.

### 2.3. Antioxidant System

Lipid peroxidation eventually leads to ferroptosis, and the antioxidant system plays a crucial role in inhibiting ferroptosis after ICH. From several angles, we shall describe the antioxidant system that controls ferroptosis.

#### 2.3.1. GPX4 and GSH in Ferroptosis

GPX4 is a selenium-dependent endogenous antioxidant enzyme that reduces toxic lipids to non-toxic lipids and inhibits lipid peroxidation [32]. A glutamic acid/cysteine antiporter called System X_c_^–^, which is located in the cytoplasmic membrane and contains solute carrier family 3 member 2 (SLC3A2) and Solute Carrier Family 7 Member 11 (SLC7A11), is responsible for transferring cysteine to the intracellular space and glutamate to the extracellular space [48]. In cells, glycine, glutamic acid, and cysteine work together to synthesize glutathione (GSH), which is involved in the synthesis of GPX4, exerting antioxidant functions [38]. To lessen the damage to cell membranes, GPX4 transforms lipid hydroperoxides to alcohol or free hydrogen peroxide, while glutathione (GSH) is transformed into oxidizing glutathione (GSSG) [49]. As a result, GPX4 and GSH are crucial in preventing ferroptosis, which is currently a hot study issue.

#### 2.3.2. FSP1–CoQ_10_–NAD (P) H Pathway

The FSP1–CoQ_10_–NAD (P) H pathway is another antioxidant mechanism that regulates ferroptosis. Recent reports have found that ferroptosis suppressor protein 1 (FSP1) can exert the same antioxidant effects as GPX4 through the FSP1–CoQ_10_–NAD (P) H pathway. NADPH, a cofactor of CoQ_10_, primarily transports electrons and hydrogen atoms. FSP1 directly converts ubiquinone (CoQ_10_) to ubiquinol, a lipophilic radical-trapping antioxidant (RTA), thereby inhibiting lipid peroxidation [41,50,51]. Although Bersuker [41] found that FSP1 expression was positively correlated with ferroptosis disease resistance in many cancer cell lines, the relationship between FSP1 and ferroptosis after ICH has not been thoroughly studied, so upregulating FSP1 may be a promising research direction for inhibiting ferroptosis after ICH.

#### 2.3.3. GCH1–BH4–DHFR Axis

Through whole-gene activation screening, a recent study found the GCH1–BH4–DHFR system to be a third independent antioxidant pathway [52]. Guanosine triphosphate cyclohydrolase 1 (GCH1) is a rate-limiting enzyme for tetrahydrobiopterin (BH4) synthesis. BH4 is a radical-trapping antioxidant that protects lipid membranes from ferroptosis. By regenerating oxidized BH4, dihydrofolate reductase (DHFR) inhibits ferroptosis [53]. In addition to acting as an antioxidant, BH4 can change the amino acid phenylalanine into the amino acid tyrosine, aiding in the production of CoQ_10_ and contributing to the FSP1–CoQ_10_–NAD(P)H pathway [52]. However, more research is required to fully understand the complex pathogenic mechanism behind ferroptosis following ICH due to a lack of understanding of the GCH1/BH4/DHFR system.

#### 2.3.4. The Mevalonate Pathway

The mevalonate pathway is a component of the FSP1–CoQ_10_–NAD (P) H pathway, which regulates CoQ_10_ and, thus, plays a role in ferroptosis regulation. The rate-limiting enzyme in the mevalonate pathway, which results in the formation of cholesterol, squalene, CoQ_10_, and prenyl pyrophosphate (IPP), is 3–hydroxy–3–methylglutaryl–CoA reductase (HMGCR) [40]. IPP can protect cells by enhancing the translation of GPX4 by stabilizing selenocysteine-specific tRNA, and it is also a precursor of squalene and CoQ_10_ [54]. Squalene has an antioxidant effect on some cancer cells, but further research is needed to determine how it affects nerve cells after ICH [55]. CoQ_10_ participates in the FSP1–CoQ_10_–NAD(P)H pathway to protect cells from ferroptosis [51]. Since cholesterol is ubiquitous in eukaryotic cells, it is also believed that it is involved in ferroptosis; however, the relationship between cholesterol oxidation and ferroptosis has not been studied. It has been reported that cholesterol, especially 7-Dehydrocholesterol (7–DHC), may be a potential regulator of lipid peroxidation and ferroptosis [40].

#### 2.3.5. The Nrf2/ARE–GPX4 Pathway

A transcription factor called nuclear factor erythroid 2–related factor 2 (Nrf2) controls glutathione production and, hence, ferroptosis. In two stages, glutamic acid, cysteine, and glycine are converted into the amino acid glutathione. In the first step, glutamate and cysteine are joined to create the dipeptide g–GluCys. Then, glycine and g–GluCys bind to create GSH. The initial and most time-consuming steps in glutathione synthesis are catalyzed by glutamic acid cysteine ligase (GCL, formerly known as glutamyl cysteine synthase). The enzyme that catalyzes the conversion of g–GluCys to GSH is known as glutathione synthase (GS). Nrf2 is a transcriptional regulator of GCL and GS. Nrf2 regulates the expression of multiple genes by binding to the Anti–Oxidative Response Element (ARE) [37]. To remain stable in non-stress conditions, Nrf2 binds primarily to the Kelch–like ECH–associated protein 1 (Keap1). Keap1 is degraded and separated from Nrf2 once oxidative stress is induced, and Nrf2’s translocation into the nucleus after binding to ARE triggers the transcription of associated genes [44]. *GPX4* and *SLC7A11* are *Nrf2* target genes; therefore, by directly increasing the transcription of *GPX4* and *SLC7A11*, *Nrf2* activation can defend cells from lipid peroxidation. Additionally, Nrf2 activation can decrease cellular iron intake, increase iron storage, and limit ROS production [23]. Therefore, it is clear that the Nrf2/ARE–GPX4 pathway plays a crucial role in controlling ferroptosis, which offers a useful route for future study on preventing ferroptosis following ICH.

### 2.4. Other Ways

#### 2.4.1. Mitochondrial Role in Ferroptosis

So far, it has been debated whether mitochondria play a role in ferroptosis. It has been discovered that glutamine is catalyzed to glutamate by glutaminase 2 (GLS2), and that glutamate is converted to α–Ketoglutaric acid (α–KG) by glutamate dehydrogenase 1 (GLUD1), which is then added to the tricarboxylic acid cycle (TAC) and electron transport chain (ETC) in mitochondria. The transaminase inhibitor aminooxyacetate (AOA) has been shown to inhibit ferroptosis in mouse embryonic fibroblasts (MEFs) [56]. The evidence suggests that mitochondria play an important role in cysteine deprivation-induced ferroptosis. Under cysteine deprivation, glutamate catabolism would promote mitochondrial respiration, generate ROS, and accelerate the depletion of GPX4 and GSH, thereby inducing ferroptosis, which further extends the mechanism of mitochondrial involvement in ferroptosis [57]. The discovery that Mito TEMPO, a mitochondrial-only superoxide scavenger, inhibits ferroptosis in cardiac cells by scavenging mitochondrial lipid peroxidation supports a key role for mitochondrial lipid peroxidation in ferroptosis [58]. Aside from that, the available evidence indicates a link between mitochondrial mechanisms and tumor cell metabolism [59], and it remains to be seen whether the same mechanisms are present in ferroptosis following ICH.

The evidence that is currently available suggests that Dihydroorotate Dehydrogenase (DHODH) can convert ubiquinone (CoQ_10_) to ubiquinol in mitochondria, preventing ferroptosis via a mitochondrial mechanism [60]. However, this study only demonstrates the promise of the DHODH mechanism in cancer therapy and further research is needed to determine its potential in ferroptosis following ICH.

#### 2.4.2. Energy Stress

Through the AMP–activated protein kinase (AMPK) pathway, energy stress can phosphorylate acetyl coenzyme A carboxylase (ACC), which reduces the biosynthesis of PUFAs and other fatty acids and inhibits ferroptosis [61]. Does this mechanism function after ICH during ferroptosis? To support it, more research is required.

## 3. Potential Interventions Targeting Ferroptosis after ICH

### 3.1. The Lipid Peroxidation Pathway

Polyunsaturated fatty acid chains (PUFAs) in cell membrane lipids undergo a sequence of events that results in the formation of lipid reactive oxygen species. Lipid hydroperoxides can generate harmful lipid radicals, such as alkoxy radicals, in the presence of iron, causing cellular damage. Moreover, these radicals can transfer protons from nearby PUFAs, triggering a new round of lipid oxidation events and causing additional oxidative damage. The accumulation of lipid ROS and the reduction in PUFAs were inhibited by the small-molecule antioxidant ferropstatin-1, thus blocking the process of ferroptosis [62]. Therefore, lipid reactive oxygen species-mediated cellular damage is required for ferroptosis.

Lipid peroxidation plays an important role in the regulation of ferroptosis, with the key enzymes ACSL4 and LOXs, as well as ROS, serving as important regulatory targets (Table 2).

#### 3.1.1. ACSL4

Lipid peroxidation is a critical step in neuronal ferroptosis, and one of the essential enzymes in lipid peroxidation is ACSL4. Recent reports have found that long non-coding RNA H19 (lncRNA H19) can promote brain microvascular endothelial cells (BMVECs) ferroptosis by modulating the miR–106b–5p/ACSL4 axis, and the knockdown of H19 increases the mRNA expression of SLC7A11 and GPX4 and downregulates the TFR1 levels [36]. The report has found that the new agent Paeonol can mediate the HOTAIR/UPF1/ACSL4 axis to inhibit ferroptosis in heme chloride-treated neuronal cells [33]. ACSL4 is one of the most important enzymes in the process of lipid peroxidation, and it is also one of the most important targets in the effort to prevent ferroptosis.

#### 3.1.2. LOXs and Its Products

LOXs is one of the key enzymes for the process of lipid peroxidation. It is possible for the peroxides that are catalyzed by lipoxygenase, also known as LOXs, to cause damage to cell membranes, mitochondria, and DNA [27]. Some authors have also suggested that LOXs and its products are also important research directions. The antioxidant N–acetylcysteine (NAC) protects neurons by inhibiting the toxic arachidonic acid production of ALOX5, but the therapeutic window of NAC is uncertain, and the mechanism by which it synergizes with Prostaglandin E2 (PGE2) has not been fully elucidated [35]. N–Hydroxy–N' –(4–n–butyl–2–methyl phenyl) –metamidine (HET0016) has also been found to inhibit the synthesis of the arachidonic acid metabolite 20–hydroxyecotetraenoic acid (20–HETE), demonstrating a protective effect after ICH [79].

#### 3.1.3. ROS

ROS has a high chemical reactivity and can cause severe damage to a cell structure, so regulating the number of reactive oxygen species in cells can inhibit ferroptosis effectively. Reveratrols encapsulated with polymer nanoparticle NPs (Res–NPs) have been reported to inhibit erastin-induced ferroptosis in HT22 mouse hippocampal cells by effectively inhibiting ROS production, which is a safer and more effective treatment for ICH injuries [70]. Ferrostatin–1 (Fer–1) has been shown to attenuate angiotensin II (Ang II)-induced inflammation and ferroptosis by inhibiting ROS levels and activating the Nrf2/HO-1 signaling pathway [80]. Fer-1 may exert long-term neuroprotective effects by reducing neurological deficits after ICH, restoring memory function, and modulating brain atrophy [62]. Therefore, it can be seen that the study of inhibition of ROS is also an important direction to reduce the ferroptosis of neurons.

### 3.2. Antioxidant System

The antioxidant system is the body’s primary mechanism for self-regulation in response to ferroptosis, and the study of its key targets has yielded certain results, which will be presented from four different perspectives: GPX4, System X_c_^–^, PPARγ, and the Nrf2/ARE–GPX4 pathway.

#### 3.2.1. GPX4

GPX4 is involved in post-ICH brain injury through its antioxidant effects, as previously reported in the literature, and upregulating GPX4 may be a potential strategy for improving brain damage caused by ICH [81]. Because GPX4 is a selenium-dependent endogenous antioxidant enzyme, GPX4-dependent ferroptosis, as well as cell death caused by excitatory toxicity or ER stress, can be effectively inhibited by the drug Se supplement [71]. Some drugs, such as Dauricine, Baicalin, Curcumin Nanoparticles (Cur–NPs), and O–dodecyl p–methylenebisphosphonic calix [4] arene micelles containing DRC (DPM) have been shown to inhibit ferroptosis by upregulating the expression of GPX4 [72,73,74,77]. Yang [73] recognized that Cur–NPs might regulate the Nrf2/HO–1 pathway to increase the GPX4 expression. However, the specific pathways by which these drugs regulate GPX4 remain a mystery. Recently, a study discovered an upstream target that regulates GPX4, methyltransferase–like 3 (METTL3), a member of the N6–methyladenosine (m6A) methyltransferase complex, whose silencing lowered the N6–methyladentine levels of GPX4 and increased the mRNA levels of GPX4, effectively inhibiting ferroptosis [39]. However, there is currently no evidence from clinical trials to support this conclusion.

#### 3.2.2. The Nrf2/ARE–GPX4 Pathway

The Nrf2/ARE–GPX4 pathway is also a potential therapeutic target, and a study has shown that Crocin can downregulate iron concentrations and upregulate GPX4 and SLC7A11 by Nrf2 nuclear translocations, reducing ferroptosis caused by ICH [78].

#### 3.2.3. PPARγ

The PPARγ pathway cooperates with the Nrf2/ARE–GPX4 pathway to play an antioxidant role [45]. Peroxisome proliferator–activated receptor gamma (PPARγ) is a nuclear receptor. After activation, PPARγ binds to the retinoid X receptor (RXR) and is transported to the nucleus, where it binds to the PPAR response element (PPRE) in the target genes to initiate transcription. Pioglitazone, as previously reported in the literature, can act as a PPARγ agonist in synergy with the Nrf2/ARE–GPX4 pathway to regulate transcription, inhibiting ferroptosis in neurons following ICH [76]. This conclusion provides a new therapeutic target for ICH—the PPARγ pathway, but more clinical studies are needed to confirm this discovery.

#### 3.2.4. SLC7A11

SLC7A11 is one of the members that make up System X_c_^–^, and Baicalin and Isorhynchophylline (IRN) were found to inhibit ferroptosis after ICH by upregulating the expression of SLC7A11 [75,77]. IRN can protect neurons from ICH-induced ferroptosis through the miR–122–5p/TP53/SLC7A11 pathway, but clinical trials are needed as evidence to support this conclusion [75]. Although Baicalin inhibits ferroptosis after ICH by upregulating GPX4 and SLC7A11, this research did not look into the effect of Baicalin inhibiting ferroptosis on other mechanisms, nor did it assess the long-term benefits of Baicalin on ICH [77].

### 3.3. Iron Metabolism

Ferroptosis is a cell death pathway that relies on iron, and the regulation of intracellular iron metabolism can effectively inhibit ferroptosis. Existing research primarily inhibits ferroptosis in two ways: lowering the concentration of iron in cells and promoting iron excretion from within cells.

#### 3.3.1. Iron Chelator

One of the causes of ferroptosis after ICH is intraneuronal iron overload, and the iron chelator Deferoxamine (DFO) can bind irons to reduce the concentration of iron in the LIP, thereby protecting neurons [63]. In a phase 2 clinical trial, deferoxamine mesylate was proven to be safe but the drug was found to not significantly improve good clinical outcomes at day 90 [82]. However, Selim [83] found that desferrioxamine mesylate seems to accelerate and help change the trajectory of recovery after ICH at day 180. This evidence suggests that future ICH trials should evaluate the results within at least 6 months, if feasible, preferably longer. Although Guo [63] demonstrated the therapeutic effect of DFO in brainstem hemorrhage, they only studied the short-term effects and did not conduct long-term studies. However, since traditional DFO therapy only addresses post-ICH iron overload and not ROS, Zhu [84] designed and manufactured a series of dual functional macromolecular nanoscavengers featuring high-density DFO units and catechol moieties, which can both reduce the iron concentration and effectively reduce the ROS levels. These findings provide a completely new strategy for the treatment of ICH. In a comparison study, it was discovered that the iron chelator clioquinol (CQ) may enhance neurological function by reducing cerebral edema and ROS production, as opposed to the other iron chelator deferiprone (DFP), which decreased the iron content in brain tissue but did not reduce cerebral edema or ROS and did not improve outcomes. This study also demonstrated that DFP increased DMT1 expression but not FPN expression, while CQ increased FPN expression but not DMT1 expression [69]. Minocycline has been shown to act as an iron chelator to alleviate brain damage in ICH-induced elderly female rats, but the mechanism is unclear [65,66]. Early clinical trials have shown that the intravenous injection of 400 mg of minocycline is safe in ICH and reaches a neuroprotective serum concentration, but subsequent clinical trials need to be further carried out [85]. A lipophilic iron chelator Pyridoxal Isonicotinoyl Hydrazone (PIH) has been reported that PIH can upregulate GPX4 and downregulate Cyclooxygenase-2 (COX–2) in ICH mice to prevent ferroptosis. The paper suggests that the regulation of ferroptosis may affect neuroinflammation, but the detailed mechanism between them needs to be further explored [64]. Mice with sepsis-induced heart damage are treated with dexrazoxane (DXZ) and Fer-1 to prevent ferroptosis. However, the iron chelator DXZ is not known to be linked to ICH [68]. Lactoferrin (Ltf) is a binding glycoprotein secreted by neutrophils, which has a much higher binding affinity with Fe^3+^ than with iron chelators. One study found that a decrease in lactoferrin after ICH in hyperglycemic mice aggravated ferroptosis, but the effect of lactoferrin on ICH needs to be further elaborated upon [67].

#### 3.3.2. FPN

FPN is a protein that transports iron to the outside of cells. MiR–124 antagonists increase FPN expression and reduce iron accumulation in older mice, implying that FPN upregulation or MiR–124 inhibition may be important in the treatment of neuronal death after ICH [42] (Table 3). Iron regulatory protein IRP2 (IREB2) has been shown to help with ferroptosis [43]. IRP2 is a direct target for MiR–19b–3p. Exosomals derived from miR–19b–3p–modified adipose–derived stem cells (ADSC–19bM–Exos) can inhibit IRP2 expression, reduce TFR1 levels, and significantly increase FPN levels. Exosome therapy in combination with miR–19b–3p may represent a promising strategy for ICH therapy [86].

## 4. Summary

In this study, we describe the impacts of lipid peroxidation, the antioxidant system, and iron metabolism on ferroptosis following ICH, which has far-reaching implications for ICH treatment and prognosis improvement. In addition to shedding light on how the disease develops following ICH, research into the mechanism of ferroptosis can suggest potential ICH treatment targets.

After ICH, iron metabolism is closely related to ferroptosis. Heme iron contributes to iron overload, and non-heme iron is also involved in the pathophysiology of ferroptosis. Studies have demonstrated that iron toxicity following ICH can increase the permeability of the BBB [2], but the specific mechanism by which non-heme iron enters brain tissue through the BBB has not been elucidated. In addition to the iron chelator DFO [82], is it possible to convert other iron chelators, such as minocycline, PIH, and lactoferrin, into clinical trials? In addition to iron chelators, the protein FPN, that regulates iron transport out of cells, can serve as a research target; this will be the focus of future studies. Lipid peroxidation is the central mechanism of ferroptosis, but the mechanism by which lipid peroxidation causes ferroptosis is still a mystery. In addition to the potential mechanism proposed by Yan [46], is there any other mechanism that can explain how lipid peroxidation leads to ferroptosis? The antioxidant pathway in ferroptosis is a physiological process in which the body spontaneously resists lipid peroxidation, and the widely known pathways are mainly GPX4 and the FSP1–CoQ_10_ –NAD(P)H pathway, but recent studies have shown that the GCH1–BH4–DHFR axis is another pathway independent of the above two pathways, but specific studies on this pathway are still lacking; therefore, the specific elucidation of this pathway is anticipated to provide novel insights [52]. In addition to the previously mentioned pathways, the pathway by which Nrf2 regulates ferroptosis via genes is also of interest, as its ability to regulate the expression of multiple genes in cells demonstrates its significance in regulating cellular metabolism [37]. As the study progressed, it was discovered that PPARγ regulates the Nrf2-associated pathway [76], a novel therapeutic target for ICH that could be the focus of future research. In recent years, a great deal of attention has been paid to the role of mitochondria in ferroptosis; however, the relationship between mitochondria in ferroptosis and ICH has not been investigated, providing a novel idea for future research [57].

## 5. Conclusion

In conclusion, research into ferroptosis after ICH will play a significant role in the treatment of ICH. By inhibiting ferroptosis after ICH, patients will experience significant benefits and the problem of high mortality after ICH will be resolved.

## Figures and Tables

**Figure 1 cells-12-00090-f001:**
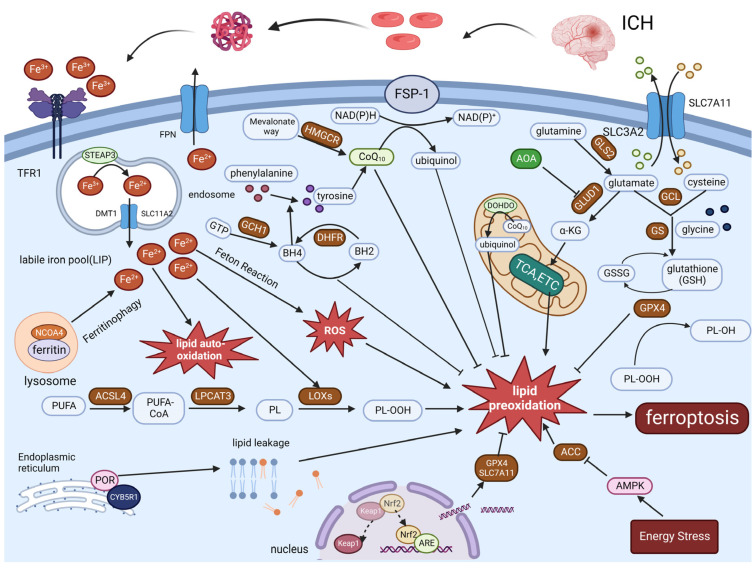
Ferroptosis is primarily composed of some components: lipid peroxidation, the antioxidant system, iron metabolism, and others. The core of ferroptosis is lipid peroxidation, and the interplay between these components regulates the occurrence of ferroptosis. Abbreviations: intracerebral hemorrhage (ICH); transferrin receptor 1(TFR1); ferroportin (FPN); divalent metal transporter 1(DMT1); labile iron pool (LIP); acyl–CoA synthetase long–chain family member 4 (ACSL4); lysophosphatidylcholine acyltransferase 3 (LPCAT3); lipoxygenases (LOXs); guanosine triphosphate cyclohydrolase 1 (GCH1); Tetrahydrobiopterin (BH4); dihydrofolic acid (BH2); dihydrofolate reductase (DHFR); 3–hydroxy–3–methylglutaryl–CoA reductase (HMGCR); Isopentenyldiphosphate (IPP); ferroptosis suppressor protein 1 (FSP–1); glutamic acid cysteine ligase (GCL); glutathione synthetase (GS); glutathione disulfide (GSSG); glutathione peroxidase 4 (GPX4); cytochrome P450 oxidoreductase (POR); NADH–cytochrome b5 reductase 1 (CYB5R1); nuclear receptor coactivator 4 (NCOA4); glutaminase 2(GLS2); aminooxyacetate (AOA); tricarboxylic acid cycle (TAC); electron transport chain (ETC); Dihydroorotate Dehydrogenase (DHODH); AMP–activated protein kinase (AMPK); acetyl coenzyme A carboxylase (ACC); α–Ketoglutaric acid (α–KG); Polyunsaturated fatty acid (PUFA); phospholipid (PL); nuclear factor erythroid 2–related factor 2 (Nrf2); Kelch–like ECH–associated protein 1 (Keap1); Anti–Oxidative Response Element (ARE); ubiquinone (CoQ_10_).

**Table 1 cells-12-00090-t001:** Genes associated with ferroptosis after intracerebral hemorrhage (ICH).

Gene	Name	Pathway	Function
*ACSL4* [31]	acyl–CoA synthetase long–chain family member 4	lipid peroxidation	catalyzes PUFA to PUFA–CoA
*lncRNA H19* [36]	Long non-coding RNA H19	lipid peroxidation	promotes BMVECs (brain microvascular endothelial cells) ferroptosis by modulating the miR–106b–5p/ACSL4 axis
*LOXs* [28]	lipoxygenases	lipid peroxidation	phospholipid (PL) is oxidized by LOXs to PL–OOH
*GPX4* [32]	glutathione peroxidase 4	GPX4 and GSH	reduces toxic lipids to non-toxic lipids and inhibits lipid peroxidation
*GCL* [37]	glutamic acid cysteine ligase	GPX4 and GSH	involved in the synthesis of glutathione
*GS* [37]	glutathione synthetase	GPX4 and GSH	involved in the synthesis of glutathione
*SLC7A11* [38]	solute carrier family 7 member 11	GPX4 and GSH	responsible for transferring cysteine and glutamate
*METTL3* [39]	methyltransferase–like 3	GPX4 and GSH	lowers N6–methyladentine levels of GPX4; increases mRNA levels of GPX4 by silencing
*HMGCR* [40]	3–hydroxy–3–methylglutaryl–CoA reductase	mevalonate pathway	rate-limiting enzyme in the mevalonate pathway
*FSP1* [41]	Ferroptosis suppressor protein 1	FSP1–CoQ_10_–NAD(P)H pathway	reduces CoQ_10_ to ubiquinol
*FPN* [42]	ferroportin	iron metabolism	transports iron out of cells
*IRP2* [43]	Iron regulatory protein 2	iron metabolism	increases TFR1 levels, and reduces FPN levels
*Nrf2* [44]	nuclear factor erythroid 2–related factor 2	Nrf2/ARE–GPX4 pathway	directly or indirectly involved in modulating ferroptosis, including metabolism of GSH, iron, and lipids, as well as mitochondrial function
*PPARγ* [45]	Peroxisome proliferator-activated receptor gamma	Nrf2/ARE–GPX4 pathway	regulates transcription with the Nrf2/ARE–GPX4 pathway

**Table 2 cells-12-00090-t002:** Drugs/molecules associated with ferroptosis after intracerebral hemorrhage (ICH).

Target	Drugs/Molecules	Influence
Iron	Deferoxamine (DFO) [63]	Iron chelator, reduces the concentration of intracellular iron
	Pyridoxal Isonicotinoyl Hydrazone(PIH) [64]	Iron chelator, reduces the concentration of intracellular iron
	Minocycline [65,66]	Iron chelator, reduces the concentration of intracellular iron
	Lactoferrin (Ltf) [67]	May prevent iron-dependent lipid peroxidation
	Dexrazoxane (DXZ) [68]	Inhibits ferroptosis in mice with sepsis-induced cardiac injury together with Ferrostatins–1 (Fer-1)
	Clioquinol (CQ) [69]	Reduces cerebral edema and ROS production
	Deferiprone (DFP) [69]	Decreases iron content in brain tissue but does not reduce cerebral edema or ROS
ROS	Ferrostatins–1 [62]	Inhibits ROS production
	Res–NPs [70]	Inhibits ROS production
GPX4	Selenium [71]	Increases antioxidant GPX4 expression
	Dauricine [72]	Inhibits ferroptosis by upregulating the expression of GPX4
	Curcumin Nanoparticles (Cur–NPs) [73]	Inhibits ferroptosis by upregulating the expression of GPX4
	DPM [74]	Inhibits ferroptosis by upregulating the expression of GPX4
CoQ_10_	Ferroptosis Suppressor Protein 1 (FSP1) [41]	Reduces CoQ_10_ to generate a lipophilic RTA that halts the propagation of lipid peroxides
ACSL4	Paeonol [33]	Mediates HOTAIR/UPF1/ACSL4 axis to inhibit ferroptosis in heme chloride-treated neuronal cells
lipoxygenase (LOXS) and its products	N–acetylcysteine(NAC) [35]	Protects neurons by inhibiting the toxic arachidonic acid production of ALOX5
SLC7A11	Isorhynchophylline (IRN) [75]	Protects neurons from ICH-induced ferroptosis through the miR–122–5p/TP53/SLC7A11 pathway
PPARγ	Pioglitazone [76]	Acts as a PPARγ agonist in synergy with the Nrf2/ARE–GPX4 pathway to regulate transcription
GPX4, SLC7A11	Baicalin [77]	Inhibits ferroptosis by upregulating the expression of GPX4 and SLC7A11
GPX4, SLC7A11, Nrf2	Crocin [78]	Downregulates iron concentrations and upregulates GPX4 and SLC7A11 by Nrf2 nuclear translocation

**Table 3 cells-12-00090-t003:** Research on the mechanism of ferroptosis after intracerebral hemorrhage (ICH).

References	In Vitro		In Vivo		Findings
	Cell	Experimental Model	Animal	Animal Model of ICH	
[87]	No*	No	12-week-old C57BL/6 male mice	Collagenase VII–S	We observed ferroptosis in the injured striatum during the acute phase of ICH
[33]	Primary cortical neurons (PCN), HT22 cells, 293T cells	Hemin, Paeonol, ferrostatin–1, Cell Transfection	C57BL/6 mice aged 8–12 weeks	Collagenase VII–S	Paeonol inhibits the progression of ICH by mediating the HOTAIR/UPF1/ACSL4 axis
[71]	PCN, HT22 cells	Hemin, L–homocysteic acid, RSL3, FIN56or erastin	Male C57BL/6 mice aged 8–12 weeks	Collagenase	Selenium increases antioxidant GPX4 expression to block ferroptosis
[42]	No	No	20-month-old C57bl/6 mice	20 μL autologous blood	The critical role of miR124/FPN-signaling in iron metabolism
[65]	No	No	18-month-old male Fischer rats	100 μL autologous arterial blood	Minocycline reduces the concentration of intracellular iron to block ferroptosis
[62]	No	No	8-week-old male ICR mice and 8-week-old male C57BL/6 mice	30 μL autologous blood	Inhibition of neuronal ferroptosis in the acute phase of ICH shows long-term cerebroprotective effects
[36]	Brain microvascular endothelial cells (BMVECs)	Hemin, Cell Transfection	No	No	Long non-coding RNA H19 protects against ICH injuries via regulating microRNA–106b–5p/acyl–CoA synthetase long-chain family member 4 axis
[66]	No	No	18-month-old female Fischer rats	100 μL autologous whole blood	Minocycline reduces the concentration of intracellular iron to block ferroptosis
[76]	No	No	Male SD rats aged 10 weeks	100 μL autologous blood	Activation of the PPARγ prevents ICH through synergistic actions with the Nrf2
[77]	PC12 cell line, PCN	Hemin, RSL3 or erastin and baicalin	C57BL/6 mice aged 10 weeks	Type IV collagenase	Baicalin enhanced the expression of GPX4 and SLC7A11 to inhibit ferroptosis in ICH
[63]	No	No	Adult male Sprague Dawley rats	Type VII collagenase	DFO reduces the concentration of intracellular iron to block ferroptosis
[79]	No	No	Adult male C57BL/6J mice	Collagenase VII–S	Lipid peroxidation was decreased and expression of GPX4 was increased to block ferroptosis
[35]	PCN	Hemin	C57BL/6 mice	Collagenase	N–acetylcysteine (NAC) protects neurons by inhibiting the toxic arachidonic acid production of ALOX5
[51]	No	No	Adult male C57BL/6 mice	30 μL autologous whole blood	DPM inhibits ferroptosis by upregulating the expression of GPX4
[88]	Primary neuronal cell	Hemoglobin	Male Sprague Dawley rats	50 μL autologous whole blood	Minocycline reduces the concentration of intracellular iron to block ferroptosis
[70]	HT22 cells	Erastin	C57BL/6 mice (8 weeks old)	Type IV collagenase	Res–NPs inhibits ROS production in ICH treatment
[73]	No	No	Male C57BL/6 mice (8–10 weeks old)	Type IV collagenase	Cur–NPs treatment might increase the GPX4 expression by regulating Nrf2/HO–1 pathway
[86]	PCN	Hemin and Cell Transfection	C57BL/6 mice (8–12 weeks)	Collagenase	ADSC–19bM–Exos inhibits IRP2 expression, reduces TFR1 levels, and significantly increase Fpn levels to block ferroptosis
[64]	PC12 cell line	PIH and erastin	Adult male C57BL/6 mice	Collagenase VII–S	PIH reduces the concentration of intracellular iron to block ferroptosis
[39]	BMVECs	Hemin, RSL3, ferrostatin–1 and Cell Transfection	C57BL/6 mice	Collagenase IV	Methyltransferase like 3 silencing effectively suppresses ferroptosis by regulating GPX4.
[81]	No	No	Male Sprague Dawley (SD) rats	100 μL autologous blood	Glutathione peroxidase 4 participates in secondary brain injury
[75]	HT–22 cells	IRN, ferric ammonium citrate (FAC), Cell Transfection	Adult male Sprague Dawley rats (SD rats) aged 11–12 weeks	Collagenase type VII	Isorhynchophylline relieves ferroptosis-induced nerve damage after ICH via miR–122–5p/TP53/SLC7A11 pathway
[72]	Human SH–SY5Y neuroblastoma cell lines	Dauricine, RSL3, Cell Transfection	Adult male C57BL/6 mice	Collagenase IV	Dauricine alleviated secondary brain injury after ICH by upregulating GPX4 expression
[78]	No	No	8-week-old C57BL/6 male mice	25 μL autologous blood	Crocin alleviates ICH-induced neuronal ferroptosis by facilitating Nrf2 nuclear translocation

*“NO” indicates no experimental data.

## Data Availability

Not applicable.

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
