# Peer review of "Ferroptosis and Iron Metabolism after Intracerebral Hemorrhage"

_cells, 2022, doi:10.3390/cells12010090_

Round 1

Reviewer 1 Report

Comparing to some mini-reviews with similar topic, in this manuscript authors were trying to summarize the most of published data about ferroptosis and iron metabolism after intracerebral hemorrhage, aiming at augment our understanding of mechanisms underlying ferroptosis post-ICH and therefore highlight ferroptosis related several pathways as targets for ICH treatments. While the manuscript was fairly clearly written and its major points are sound, there are several issues that need to be fixed that would significantly bolster it.

Majors:

1.     Some presentations were logical confusion. For example, while discussing the GPX4 pathway, the cascade of GPX4 (3.2.1), Nrf2/ARE–GPX4 (3.2.4), PPARγ (3.2.3), and SLC7A11 (3.2.2) sounds more logical than the one author presented.  

2.     Although plenty of published papers showed positive result of ferroptosis inhibition via iron chelator on ICH, there was a few negative reports. This needs be addressed as well.  

3.     Not all iron chelators related to ICH treatment were discussed.

Minors:

1.     Some abbreviations had no full name. For example: RCTs in line 30.

2.     There were some formation problem, such as no space between sentences (line 200, 212) and abnormal big gap within sentence (line 117).

3.      Some description was confusion. For example: “FSP1 directly reduces CoQ10 to ubiquinol (line 200)” What does “reduces” means?

4.     Grammar needs to be carefully checked.  

Reviewer 2 Report

The review by Sun et al., “Ferroptosis and Iron Metabolism after Intracerebral Hemorrhage” has summarized the mechanism of ferroptosis after intracerebral hemorrhage, focusing on lipid peroxidation, the antioxidant system, iron metabolism, and other pathways. In addition, elaborates the regulatory targets and drug molecules for the prevention and treatment of ICH. The paper was nicely written, and a few minor comments needed to consider before publication.

A few concerns/comments needed to be explained/modified:

1.     line 28: This paragraph should begin with an explanation of the primary and secondary injuries following ICH.

2.     Line 56-57: “Recent evidence suggests that ferroptosis is also involved in neurological diseases such as stroke, and degenerative diseases, especially ICH.” The expression is inaccurate and needs to be addressed.

3.     Line77: “iron is absorbed mostly from the diet” which form? Fe2+ or Fe3+?

4.     Line 95: “Microglia generally clear hematomas after ICH through three pathways:” Hematomas are cleared by microglia only? What about macrophages and astrocytes?

5.     Line 117: Please adjust the format.

6.     Line 280-281: This section needs to be expanded.

7.     Line 314-315: “Ferrostatin-1 (Fer-1) has been shown to inhibit the production of ROS and exert long-term brain protection.” Such information is imprecise and will be unclear for a reader. More details please.

8.     Line 415: “After the failure of 2 clinical trials for the iron 415 chelator DFO” The description "failure" is very imprecise. Please correct this sentence. (PMID: 35306827, 34789008 and more)

9.     It is necessary to make additions to the references. For example: line 188: “SLC3A2 and SLC7A11”; line 31-33: “INTERACT1, 2, ATACH–I, I, FAST and TICH”; line 126-127; line 214-216, etc. Please check the full text.

Round 2

Reviewer 1 Report

All the questions have been fully addressed by Authors with adequate data in details.